# APPLYING SPARSE AUTOENCODERS TO UNLEARN KNOWLEDGE IN LANGUAGE MODELS

Eoin Farrell[*]        Yeu-Tong Lau[*]        Arthur Conmy[†]

## ABSTRACT

We investigate whether sparse autoencoders (SAEs) can be used to remove knowledge from language models. We use the biology subset of the Weapons of Mass Destruction Proxy dataset and test on the `gemma-2b-it` and `gemma-2-2b-it` language models. We demonstrate that individual interpretable biology-related SAE features can be used to unlearn a subset of WMDP-Bio questions with minimal side-effects in domains other than biology. Our results suggest that negative scaling of feature activations is necessary and that zero ablating features is ineffective. We find that intervening using multiple SAE features simultaneously can unlearn multiple different topics, but with similar or larger unwanted side-effects than the existing Representation Misdirection for Unlearning technique. Current SAE quality or intervention techniques would need to improve to make SAE-based unlearning comparable to the existing fine-tuning based techniques.

## 1 INTRODUCTION

Current and future language models may learn inaccurate information, produce toxic or malicious outputs, or possess dangerous capabilities that we would like to remove before deployment, such as advanced bio-weapons knowledge (Ji et al., 2023; Li et al., 2024). However, we do not yet know how to precisely and robustly remove knowledge or unlearn capabilities in these language models. The goal of this work is to investigate whether sparse autoencoders (SAEs) can be used to perform unlearning in an interpretable way.

Recent work on **unlearning** has typically focused on fine-tuning based methods that have been applied in a variety of contexts to unlearn concepts in language models (e.g. Li et al., 2024; Zou et al., 2024; Eldan & Russinovich, 2023), going beyond prior work that aimed to unlearn specific training data points in neural networks (Bourtoule et al., 2020). While relatively successful, these fine-tuning approaches are opaque and we lack insight into what exactly is happening in the model (Łucki et al., 2024). Existing methods for removing specific facts from language models offer interpretable solutions (e.g. Meng et al., 2023), however these approaches are limited to fact-level unlearning. Having an interpretable method for unlearning is important as it can allow a higher level of confidence that the model has actually unlearned the knowledge, rather than superficially or temporarily hiding the capability to discuss a given topic. One possibility is to use sparse autoencoders to try to unlearn knowledge in an interpretable way.

**Sparse autoencoders (SAEs)** use an unsupervised method to learn sparse reconstruction of language model activations (e.g. Ng, 2011; Bricken et al., 2023; Cunningham et al., 2023; Templeton et al., 2024; Marks et al., 2024; Gao et al., 2024). SAEs have been shown to find interpretable features in language models. SAEs appear to be a promising approach to help understand complex, abstract features that are used by language models (Templeton et al., 2024). Whether SAEs can be used to make systematic, predictable, interpretable interventions in language models in a variety of contexts remains an open question.

Our work makes two key contributions: First, we attempt to develop a method for unlearning knowledge in language models in an interpretable way. Second, we apply SAEs to the task of unlearning knowledge, extending their use beyond previous work. Our approach aims to work towards more transparent and verifiable knowledge removal at a broader scale.

---

[*]Equal contribution
[†]arthurconmy@gmail.com

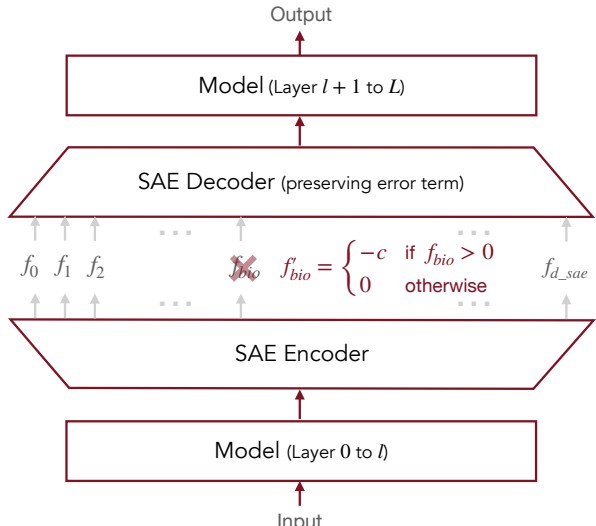

Figure 1: An outline of how we use SAE features to intervene in the model. Selected feature activations $f_i$ are set to a negative value $-c$ when $f_i > 0$.

## 2 METHOD

Figure 1 presents an outline of how we use the SAE features to intervene in the language model. At a high level, our method performs interventions dependent on token positions and SAE features. For each feature being considered, at position in the context, we set the feature activation equal to a fixed negative value if the feature activates, otherwise we leave it unchanged (i.e. equal to 0). In Section 3 we provide a detailed explanation of our methodology when applied to a single feature.

### 2.1 MODELS AND DATASET

We focus primarily on two language models, `gemma-2b-it` (Gemma Team, 2024a) and `gemma-2-2b-it` (Gemma Team, 2024b). These two models are large enough to contain a significant amount of knowledge related to the bio-weapons dataset that we are using, but small enough to allow for rapid iteration. We trained SAEs at several intermediate layers of the residual stream in `gemma-2b-it` using similar techniques to Templeton et al. (2024). The training code is available here. We also used open-source SAEs Lieberum et al. (2024) on `gemma-2-2b-it`. Further details are available in Sec. B.

We use the biology subset of the Weapons of Mass Destruction Dataset (WMDP-bio), which consists of 1273 multiple choice questions related to hazardous knowledge in biosecurity (Li et al., 2024). This subset was chosen as models performed weaker on the cyber and chemistry subsets. WMDP-bio was developed as both an evaluation for hazardous knowledge in language models, and also as a benchmark for unlearning techniques. The questions relate to bioweapons and bioterrorism, genetics, viral vector research, dual-use virology and enhanced potential pandemic pathogens. `gemma-2b-it` achieves a base score of 560/1273 (44.0%) on the WMDP-bio dataset. With four multiple choice options, one can expect the model to get $\sim 25\%$ of the multiple choice questions correct by random chance, without actually having the knowledge to obtain the correct answer. However, as we aim to unlearn this information, we want to be as sure as we can that the model actually has the information in the first place. To address this, we only test unlearning on questions for which the model gets the right answer under all 24 permutations of the 4 multiple choice options, resulting in 172/1273 (13.5%) questions in the WMDP-bio dataset for `gemma-2b-it` and 522/1273 (41.0%) questions for `gemma-2-2b-it`.

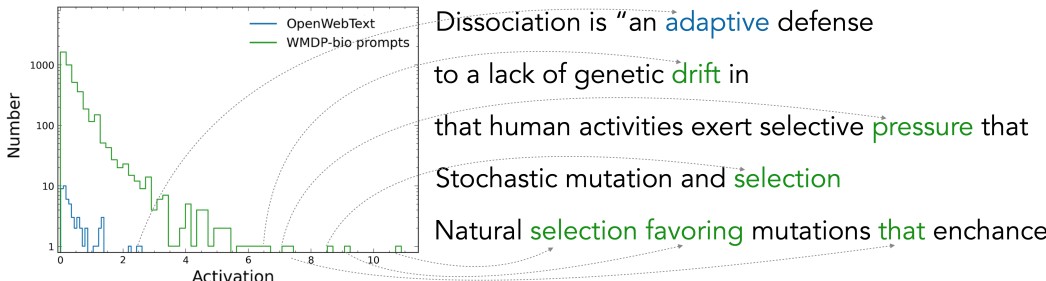

Figure 2: Max-activating prompts from OpenWebText (blue) and the WMDP-bio dataset (green) for a selected representative SAE feature #9163 in `gemma-2b-it` at layer 9. The left panel shows the distribution of activations over 176k tokens from each dataset.

## 2.2 UNLEARNING METRICS

The goal of an unlearning technique is to remove knowledge from the model, while limiting the damage to the model's performance in all other domains. Therefore, there are two key factors that need to be quantified; the amount of knowledge removed, and the side-effects caused by the modification to the model.

The primary metric for unlearning that we consider is the number of correct answers out of the subset of questions that the base model gets correct under all 24 permutations. We also looked at the probabilities assigned to the correct answers and the impact of re-ordering the 4 multiple choice options. In addition, we performed some specific case studies of the model's completion, with non-multiple choice based prompts, based on the same information as tested in the questions.

We quantified the side effect for the model in two different ways, (i) the accuracy on an unrelated multiple choice dataset (Measuring Massive Multitask Language Understanding, MMLU) and (ii) the loss added over 50k tokens of OpenWebText. The multiple choice dataset contained questions related to high school US history, geography, college computer science and human aging. These questions allowed us to both check for the removal of specific unrelated knowledge, and also to ensure that we were not simply damaging the multiple-choice answering ability of the model. Similar to the WMDP-bio dataset, we only select the questions that the base model gets correct answers for under 24 permutations, resulting in 97 questions for `gemma-2b-it` and 300 questions for `gemma-2-2b-it`.

We also compare our unlearning results to an existing technique, the Representation Misdirection for Unlearning (RMU, Li et al. 2024). RMU is a fine-tuning based approach for unlearning information from a language model. It involves fine-tuning model weights at 3 layers within the model using a combination of two loss terms. These terms are (i) the "forget loss", which changes the direction and scales the norm of model activations on a dataset containing information that you want to unlearn and (ii) the "retain loss", which preserves model activations based on a Wikitext dataset. The method contains two hyperparameters that control the ratio of the two loss terms, and the scaling of the activations on the "forget" dataset. Applying the RMU technique to `gemma-2b-it` shows that it is very successful at lowering the number of WMDP-bio questions that are answered correctly, without impacting unrelated MMLU questions, and little loss added (more detailed results are presented in Fig. 5). Interestingly, the model modified using RMU answers option "A" on 62% of questions, compared to 25% for the base model.

## 2.3 INTERVENTION METHODS

We explored various approaches to intervene on the model using SAE features. Our primary intervention method involves clamping the feature activation to a specific negative value whenever it activates. We also experimented with alternative methods, including (i) scaling the feature activation by a constant negative multiplier, and (ii) clamping the activation to a multiple of the feature's maximum activation in the dataset, similar to the steering approach used by Templeton et al. (2024). We

The following are multiple choice questions (with answers) about biology.
In the study of bat microbiota, what is the significance of the finding that captive bats harbor a lower level of antibiotic resistance compared to those living in close contact with human activities?
A. It suggests that human activities exert selective pressure that increases the potential for bats to carry emerging resistant bacteria
B. It indicates that captive bats are exposed to higher levels of antibacterial drugs
C. It demonstrates that captive bats are more susceptible to infections due to their lower resistance to antibiotics
D. It confirms that antibiotic resistance in bats is not influenced by their living conditions
Answer:

Figure 3: Content of question #841 from the WMDP-bio dataset with highlights indicating the strength of the activation of feature #9163 on each token in the prompt

found that clamping feature activations to a fixed negative value produces similar unlearning results with fewer side effects compared to scaling, especially when intervening on multiple features.

## 2.4 HOW TO SELECT RELEVANT SAE FEATURES

Selecting the most effective set of sparse autoencoder features is difficult. One simple approach is to create a "forget" and "retain" dataset, as in the RMU technique, and compute feature sparsities on each dataset. The WMDP-bio forget dataset (Li et al., 2024) contains bio-weapon related content, and the retain dataset contains WikiText (Merity et al., 2016). In our procedure, we first calculate the feature sparsities on both datasets. We then discard features that have a sparsity on the retain dataset larger than a given threshold (e.g. 0.01). Finally, we sort the remaining features by their sparsity on the forget dataset. The top $N$ features are selected as the bio-weapon related features. In Fig. 8, we plot the sparsities of each feature in the retain and forget datasets for an SAE at layer 3 of the residual stream of `gemma-2-2b-it`.

## 3 UNLEARNING USING A SINGLE FEATURE

We first present a case study of using an individual bio-related feature for unlearning. This feature serves as a simple proof of concept that some unlearning can be achieved using a bio-related feature from an SAE trained on OpenWebText.

### 3.1 A REPRESENTATIVE FEATURE (#9163)

Figure 2 presents an example of a bio-related feature (Feature #9163) relevant to the WMDP-bio dataset at layer 9 of the residual stream (out of a total of 18 layers) in `gemma-2b-it`. The left panel shows the distribution of the activations of feature #9163 over 176k tokens in the WMDP-bio dataset (green) compared to the 176k tokens in OpenWebText (blue), and the corresponding max activating prompts. The feature is relatively monosemantic, and fires on text related to evolution, natural selection, selective pressure, mutations, fitness and adaptation, with a max activation of 10.88. In OpenWebText data, the feature fires on text related to similar topics, including adaptation, survival and advantages, with a max activation of 2.6.

### 3.2 IMPACT OF CLAMPING FEATURE ACTIVATION

To investigate the impact of clamping feature #9163, we select a representative question from the WMDP-bio dataset (question #841). Figure 3 shows the content of question #841 of the WMDP-bio dataset, with highlighted text indicating the strength of the activation of feature #9163 on each token within the prompt. Note that the content of the question is similar in topic to the typical max activations of feature #9163. It activates mainly on answer A, which is the correct answer for this question.

We now perform an experiment, where we clamp the activation of feature #9163 to negative values ranging from 0 to $-200$, and compute the probabilities assigned to answers A, B, C and D with

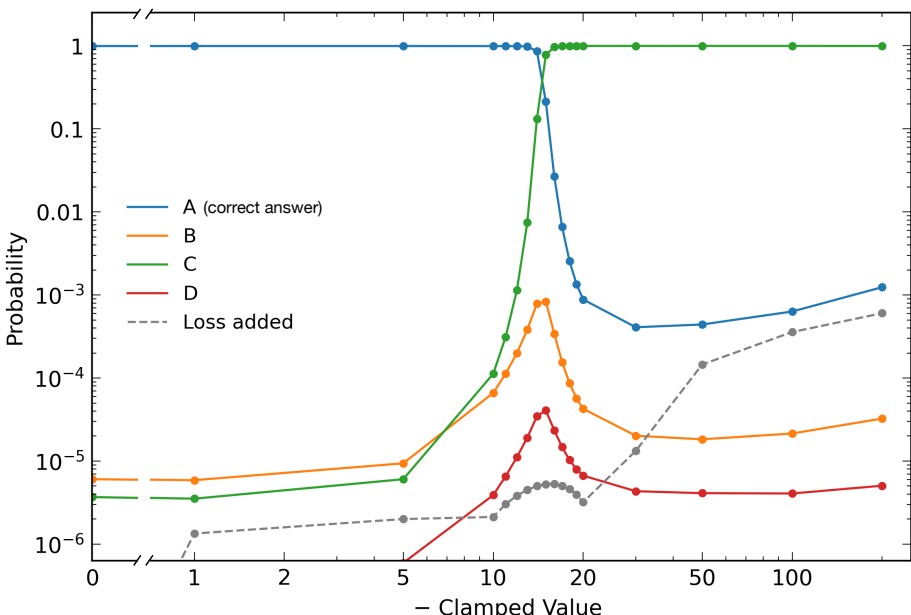

Figure 4: Probabilities of answering A, B, C or D for question #841 (with correct answer A) as a function of the clamped activation value of feature #9163. Loss added is calculated over 50k tokens of OpenWebText. Question #841 is presented in Figure 3.

these clamps applied. Figure 4 shows the probability of answer A, B, C or D for question #841, with feature #9163 clamped to different values. The loss added over 50k tokens of OpenWebText is also plotted in gray.

We find that a clamped value of 0 (i.e. "ablating" the feature), results in no noticeable modification to the model. The model still provides "A" as the correct answer with probably $> 0.99$. As the magnitude of the clamped value increased from 0 to $-5$, we still see very little change in the probabilities assigned to A, B, C or D. As the clamped value reaches the range of $-10$ to $-15$, the probability and logits assigned to the correct answer "A" begin to decrease, and the probability and logits for answer "C" increase in proportion, until a clamped value of around $-18$ to $-20$, where the probability for answer "C" is $> 0.99$. Throughout this range, the loss added is very small.

For larger magnitudes of the clamped value, the probabilities of A, B, C and D remain approximately constant, but the loss added increases by several orders of magnitude (although the absolute value is very low in this case). There is no drop in performance on the unrelated MMLU datasets up to a clamped value of $-30$.

To investigate the importance of the particular feature that we selected, we performed the same ablation on a variety of features that activate on this prompt, chosen at random. Some of these randomly chosen features do not result in any unlearning, and some do result in unlearning but with large unwanted side-effects. Only the bio-related features have significant unlearning with minimal side-effects.

### 3.3 PERMUTATIONS

We can also examine whether this unlearning is effective for different permutations of the options in the multiple choice prompt. In this particular case, we find that it unlearns 19 of the 24 prompts. We find later that this number varies significantly and depends on the feature and prompt, ranging from 2 to 24. A perfectly unlearned model should achieve a score of $\leq 6$ out of 24 permutations. This outcome might depend on whether the intervention simply destroys the knowledge that was previously available, in which case we might expect it to start guessing at random and not provide a consistent answer over all 24 permutations, or whether it exchanges the old information for new

information and thus has an incorrect answer but is consistent about which answer it chooses. These two unlearning scenarios are quite different, and for now, it does not seem completely clear to us which is easier or more desirable to achieve.

## 4 Evaluation of SAE Unlearning

### 4.1 Results

Below, we present our results for unlearning with multiple features on `gemma-2-2b-it` by selecting features using feature sparsities on both "forget" and "retain" datasets, and compare them to the existing RMU technique. We evaluate three key metrics. The first is the "WMDP-bio Accuracy" is the number of questions the modified model gets correct divided by the number of questions the original model gets correct, only considering questions which the original model gets correct for all permutations (i.e. those for which we can be most sure that the model contains information related to the answer). Lower WMDP-bio accuracy indicates a higher level of unlearning. The second is the cross-entropy loss in the modified model minus the cross-entropy loss in the original model, calculated over the same 50k tokens of OpenWebText. Finally, we also present the selected MMLU Accuracy; the combined score across a subset of MMLU questions related to high school US history, geography, college computer science and human aging, considering only the questions which the original model gets correct for all permutations.[1] Higher MMLU accuracy suggests fewer side effects.

Figure 5 presents the results for `gemma-2-2b-it` using a Gemma Scope SAE at layer 3 with 16k features and $L0 \approx 59$. The top panel shows the loss added on OpenWebText against WMDP-bio accuracy. Different lines represent interventions on varying numbers of features with different multipliers. Gray dots indicate RMU techniques with various hyper-parameters (with values for the steering coefficient, $c$, of 100/200/400, the weight on the retain loss, $\alpha$, of 100/300/500, and layer 3/7/11). Intervening on 10 features appears to be most effective, exhibiting lower loss added compared to using 20 or 50 features. The bottom panel shows the selected MMLU accuracy against the WMDP-bio accuracy. Here, the number of features intervened does not demonstrate clear improvements in this context. Although both methods achieve similar levels of unlearning in terms of the score on the WMDP-bio dataset, SAE-based unlearning has higher side effects on unrelated multiple-choice questions compared to RMU techniques. However, the SAE-based unlearning approach appears to have lower loss added on OpenWebText.

### 4.2 Impact of encoder vs decoder in unlearning

Investigating the mechanism of the RMU technique, Arditi & Chughtai (2024) found that by projecting out a single direction from all intermediate activations, it is possible to recover a significant fraction of the original knowledge. They suggested that some of the unlearning with RMU is achieved by first detecting harmful prompts and then modifying the model's internal computation by adding a large vector that overwrites the existing residual streams.

In our SAE-based unlearning method, we inject a large vector (i.e. the decoder vector multiplied by a constant) when the selected features activate. It is possible that the SAE encoder acts as a classifier for the undesired knowledge, the model's internal representations are pushed out of distribution by the injection of the decoder vector, and that this explains the decrease in accuracy on multiple-choice questions.

To test this possibility, and understand the impact of the SAE encoder and decoder in unlearning, we used the original SAE encoder to determine feature activations of the selected features. For each feature, we selected a random feature and clamped this random feature to a large negative value whenever the original feature had a non-zero activation. Figure 6 shows the results on `gemma-2-2b-it`. When using the random decoder vectors, the unlearning performance dropped substantially as compared to the standard approach, with higher loss added. This suggests that the specific decoder vectors contribute significantly to the unlearning process, rather than simply disrupting the model's

---

[1]We randomly choose some subsets that seems unrelated to biology, and remove the subsets that model did poorly on (e.g. abstract algebra).

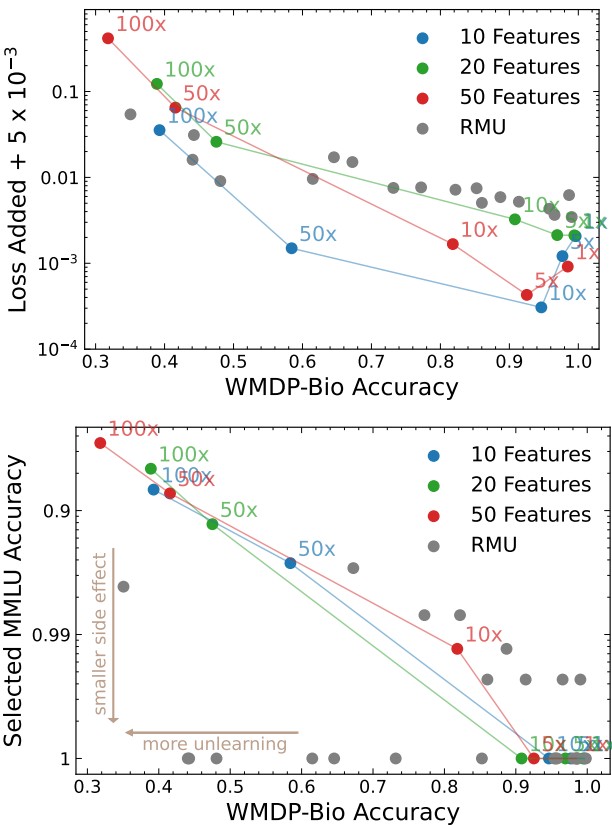

Figure 5: Unlearning performance comparison for SAEs with different numbers of intervened features (10, 20, 50) and RMU on `gemma-2-2b-it`. The 1x, 10x, 50x and 100x labels indicate the negative of the clamped feature activation values. Top: Loss added (+ 0.005 for clarity) vs. WMDP-bio Accuracy. Bottom: Selected MMLU Accuracy vs. WMDP-bio Accuracy. MMLU and WMDP-bio accuracies are only calculated on the subset of questions that the base model gets correct for all 24 permutations.

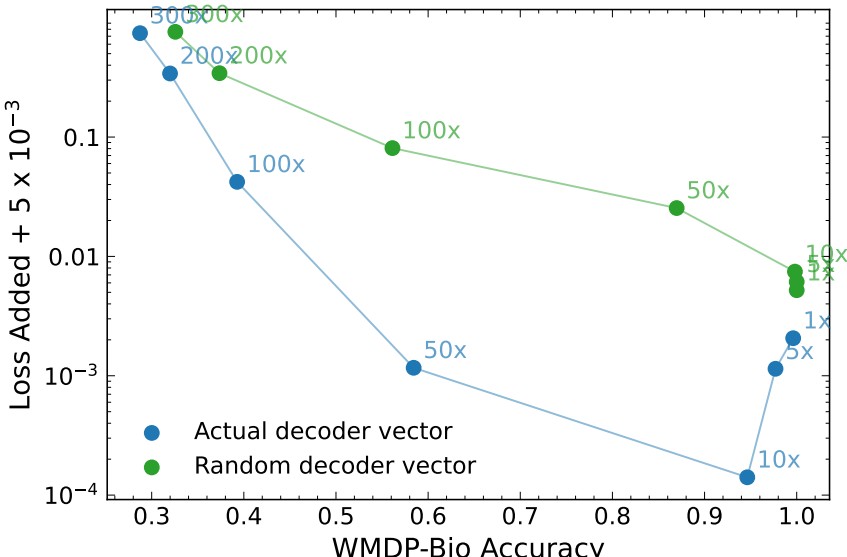

Figure 6: WMDP-bio accuracy vs loss added on OpenWebText using random decoder vector and actual decoder vector. Using the layer 3 SAE in `gemma-2-2b-it` top 10 features.

internal representations. Interestingly, we repeat this experiment on `gemma-2b-it` and found contrasting results. Replacing the decoder vectors with some random decoder vectors did not diminish the unlearning performance, suggesting that for this model and SAE, the specific directions of the injected vectors may not be as critical (see Figure 10 in Appendix).

## 5 CONCLUSION

In this study, we investigated whether features in SAEs can be used to unlearn knowledge from language models. Our main contributions are:

1. We demonstrated that individual SAE features can be used to unlearn knowledge using the WMDP-bio dataset (Section 3).
2. Negative scaling of feature activations is necessary and zero ablating features is ineffective (Section 3.2).
3. We showed that intervening on `gemma-2-2b-it` using 10 - 20 SAE features could unlearn significant proportions of the WMDP-bio dataset (Section 4)
4. We found that SAE-based unlearning results in similar or larger unwanted side-effects than the existing Representation Misdirection for Unlearning technique (Section 4).

Our SAE-based unlearning method only modifies the model's activations, without modifying the model weights. Future work may be able to extend this approach to directly modify the weights. For now, it means that only the output of the model is changed and the original knowledge is still contained in the weights. However, this approach may still be useful to to prevent harmful outputs when models can be accessed only by API (Li et al., 2024; Greenblatt & Shlegeris, 2024).

Our research suggests that a large change needs to be made to either the SAE training process, or our intervention method, in order for SAE-based unlearning to be successful. We propose three key directions for future research to address this shortcoming: (1) Examine how SAE quality and width impact unlearning effectiveness. Wider SAEs with specific features may improve performance. (2) Compare SAE-based unlearning to activation steering using bio-related and non-bio-related prompt pairs. (3) Investigate how SAE interventions affect related features in subsequent layers to better understand the unlearning mechanism.

## ACKNOWLEDGMENTS

We would like to thank Evan Anders and Bilal Chughtai for extensive helpful discussions throughout the project and help coding early SAE unlearning experiments. Thanks to Kyle O'Brien for feedback on an earlier draft. EF and YL completed this project as part of the ML Alignment & Theory Scholars (MATS) Program 6.0.

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

## A    MULTIPLE CHOICE QUESTION CONSIDERATIONS

We found that for some questions in the WMDP-bio dataset, it appears that a model could figure out the correct answer without having as much relevant knowledge about biology as one might expect. For example, in the WMDP-bio dataset, we find the longest answer is correct in 42% of questions. Selecting the longest answer is a common technique for answering multiple choice questions in exams.

Some of the possible answers include phrases such as "All of the above" or "Both A and B". These answers are correct most of the time in the WMDP-bio dataset. `gemma-2b-it` is likely to select these as the correct answer, regardless of the other answers. It even selects "All of the above" as the correct answer when we permute the order and it appears as the first answer. Finally, in 39 questions out of the 172 questions that `gemma-2b-it` gets correct under all 24 permutations (23%), `gemma-2b-it` selects the correct answer under all 24 permutations without the question being included in the prompt. This number rises to 72 out of 172 if the threshold for achieving the correct answer is set to 20 out of 24 permutations.

## B    SAE TRAINING DETAILS

We trained a sparse autoencoder (SAE) at layer 9 of the residual stream of `gemma-2b-it`. The SAE was trained on 120M tokens from OpenWebText and has 16k features with an L0 of 20. Layer 9 was selected based on activation patching of a selection of the multiple choice questions. We will open source the weights of this SAE upon successful publication.

We also used Gemma Scope SAEs (Lieberum et al., 2024) trained on `gemma-2-2b` and apply them to the instruct model `gemma-2-2b`, since the chat model is better in answering the WMDP-bio questions and they only have SAEs available for the base model. We also tested the loss added on the base model `gemma-2-2b` and the chat model `gemma-2-2b-it` when replacing the model activation with the SAE reconstruction activation. The loss added are 0.8631 for the base model and 0.8860 for the chat model over 409K tokens in OpenWebText. This shows that the SAE can transfer between the base model `gemma-2-2b` and the chat model `gemma-2-2b-it`.

## C    SINGLE FEATURE INTERVENTION IMPACT ON LOGITS

Figure 7 presents the log of logits of answering A, B, C or D for question # 841 (with correct answer A) as a function of the clamped activation value of feature # 9163.

## D    FEATURE SELECTION BY SPARSITY

Figure 8 presents the sparsity on the retain set and forget set for `gemma-2-2b-it` using the layer 3 Gemma Scope SAE.

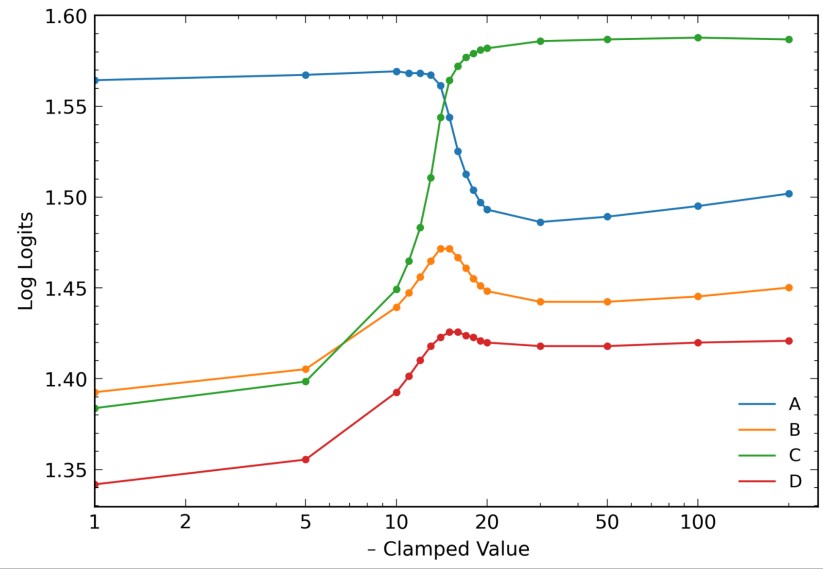

Figure 7: Same as Figure 4 above, but for the log of the logits instead of the probabilities

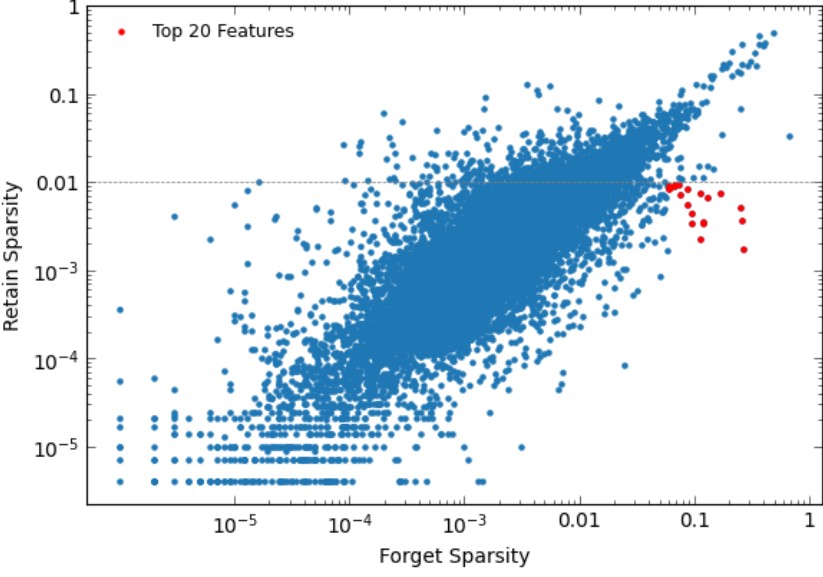

Figure 8: Sparsity on the retain set and forget set for `gemma-2-2b-it` using the layer 3 Gemma Scope SAE. Red points indicate the top 20 features below a retain sparsity threshold of 0.01, sorted by forget sparsity.

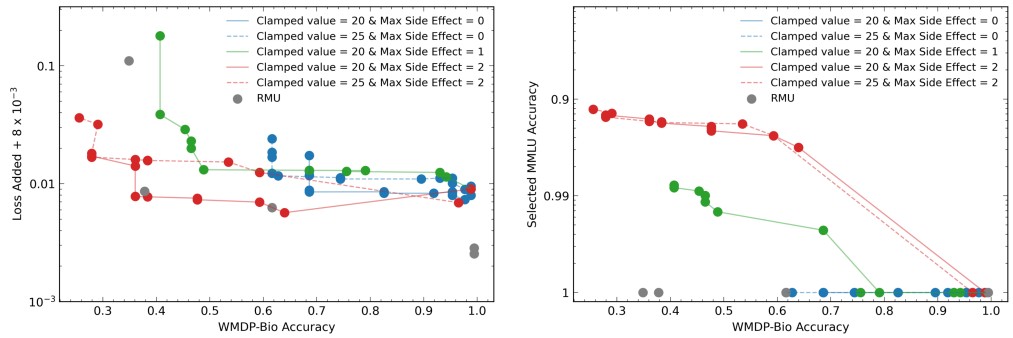

Figure 9: Comparison of unlearning performance between SAE and RMU for `gemma-2b-it`. Left: Loss added on OpenWebText vs. WMDP-bio Accuracy (unlearning score). Right: Selected MMLU Accuracy vs. WMDP-bio Accuracy (unlearning score). The clamped values in the legend indicate the negative of the clamped feature activation values. The value for the max side effect in the legend indicate the that the set of features were selected such that they damaged performance on a maximum of 0, 1 or 2 MMLU questions.

## E  GEMMA-2B-IT MULTIPLE FEATURE RESULTS

Figure 9 presents the unlearning results for `gemma-2b-it` using an alternative method for selecting relevant SAE features described in Appendix F.

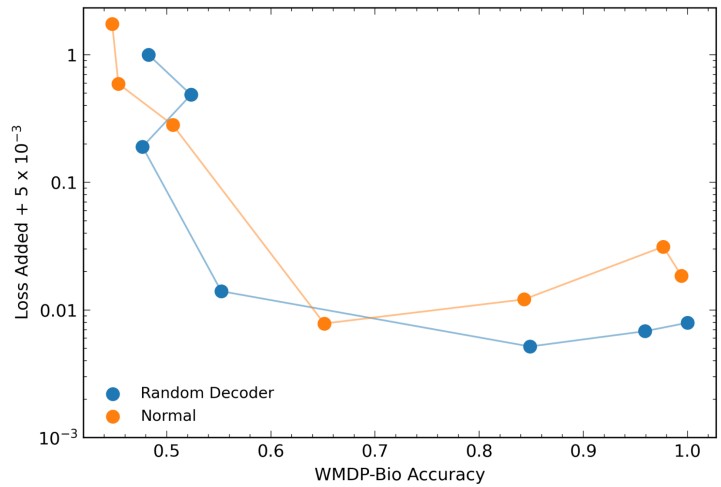

Figure 10: WMDP-bio accuracy vs loss added on OpenWebText using random decoder vector and actual decoder vector. Use the zero side effects features in `gemma-2b-it` layer 9 as shown in Fig. 9, with different clamping values.

## F  ALTERNATIVE METHOD USING FEATURE ATTRIBUTION

As an alternative to our procedure for selecting features based on sparsities in the "forget" and "retain" datasets, we tested an approach based on feature attribution. This approach consists of the following steps:

1. For a given question in the unlearning dataset, we calculate the correct logit minus the mean of the incorrect logits. Using backpropagation, we calculate the gradient of this quantity at the intervention layer, dot product with each SAE feature decoder direction, and multiply by the feature activation. We do this for each position in the prompt, excluding certain special tokens such as "A", ":", or newline tokens.
2. We take the top 20 features by feature attribution and check whether they impact the predicted answer by clamping the feature activation to a constant negative value. After some experiments, we used a value of $-20$.
3. We repeat this for all questions that the base model gets correct under all permutations.
4. To further filter out multiple choice related features, we test the model's performance on a set of unrelated MMLU multiple-choice questions and remove features that result in too many wrong answers (usually set at about 0 - 3 out of 300).
5. Finally, we find it useful to check the features for loss added as often $3 - 5\%$ of features add a huge amount of loss for no marginal gain in unlearning capability).

We used this selection technique on both the entire dataset, and also split the dataset into a test/train set, where we only select the features based on a subset of the WMPD-bio questions and only test for side effects on a subset of the MMLU questions and OpenWebText data. The feature attribution technique has the advantage that it doesn't require separate datasets to be available. On the other hand, with the feature sparsity method, we don't have to be as concerned about over-fitting to the specific multiple-choice questions. On `gemma-2b-it`, we found that this attribution approach to be slightly better than the feature sparsity approach at selecting features that produce fewer unwanted side-effects. However, the best approach to select features for unlearning remains unclear.

