# OpenReview forum: "Applying Sparse Autoencoders to Unlearn Knowledge in Language Models"
_NeurIPS.cc/2024/Workshop/SafeGenAi — SafeGenAi Poster_

### Official Review · Reviewer_Un28 · 2024-10-08
**An Interesting Exploration of SAEs for Unlearning**

**Rating:** 6
**Confidence:** 3

**Review:**

This paper investigates the use of sparse autoencoders (SAEs) for unlearning knowledge in language models, focusing on bioweapon-related information. The authors test their approach on gemma-2b-it and gemma-2-2b-it models using the biology subset of the Weapons of Mass Destruction Proxy dataset. They compare their SAE-based method to an existing fine-tuning technique called Representation Misdirection for Unlearning (RMU).

The work addresses an important problem in AI safety and ethics. As language models become more capable, the ability to selectively remove potentially dangerous knowledge becomes crucial. The authors' approach of using interpretable SAE features for unlearning is novel and shows some promise.

Pros:

(1) The paper tackles a relevant and timely issue in AI safety.

(2) The use of SAEs for unlearning is an innovative approach that could potentially offer more interpretability than existing methods.

(3) The authors provide a detailed analysis of how individual SAE features can be used to unlearn specific knowledge, demonstrating the potential for fine-grained control.

(4) The comparison with the RMU technique provides useful context for evaluating the effectiveness of the SAE-based approach.

Cons:

(1) The effectiveness of the SAE-based unlearning seems limited compared to the RMU technique, especially in terms of unwanted side-effects on unrelated tasks.

(2) The paper lacks a thorough discussion of why the SAE approach underperforms compared to RMU. More insight into the underlying reasons would be valuable.

(3) The evaluation is limited to two specific models and one dataset. A broader range of models and datasets would strengthen the conclusions.

(4) The paper doesn't adequately address the potential risks of their approach. For instance, could this method be misused to selectively remove ethical constraints from language models?

(5) The authors mention that "a large change needs to be made to either the SAE training process, or our intervention method," but don't provide concrete suggestions for such improvements.

Overall, this paper presents an interesting exploration of using SAEs for unlearning in language models. While the results are somewhat disappointing compared to existing methods, the approach is novel and could potentially lead to more interpretable unlearning techniques in the future. The work provides a useful starting point for further research in this direction.

However, the limited effectiveness of the method and the lack of concrete suggestions for improvement make it difficult to see immediate practical applications. The paper would benefit from a more in-depth analysis of why the SAE approach struggles and clearer directions for future work.

---

### Official Review · Reviewer_hM3m · 2024-10-09
**Interesting research**

**Rating:** 6
**Confidence:** 1

**Review:**

1. There seems to be a lot of whitespace in Page 7.
2. The figures in Figure 4 are unclear in grayscale.
3. Why should a perfectly unlearned model achieve a score of less than 6?